# Relationship between Skeletal Muscle Thickness and Physical Activity in 4- to 6-Year-Olds in Japan

**DOI:** 10.3390/children10030455

**Published:** 2023-02-25

**Authors:** Pengyu Deng, Hayao Ozaki, Toshiharu Natsume, Yoshihiko Ishihara, Dandan Ke, Koya Suzuki, Hisashi Naito

**Affiliations:** 1Faculty of Health and Sports Science, Juntendo University, Inzai 270-1695, Chiba, Japan; 2Institute of Health and Sports Science & Medicine, Juntendo University, Inzai 270-1695, Chiba, Japan; 3School of Sport and Health Science, Tokai Gakuen University, Miyoshi 470-0207, Aichi, Japan; 4School of Medicine, Tokai University, Isehara 259-1193, Kanagawa, Japan; 5School of Science and Technology for Future Life, Tokyo Denki University, Adachi, Tokyo 120-8551, Japan; 6School of Public Health, Fudan University, Shanghai 200433, China

**Keywords:** accelerometer, muscle thickness, growth, development

## Abstract

Purpose: Physical activity (PA) is widely recognized as a key factor in promoting skeletal muscle growth, though little is known about the specific impact of PA on the skeletal muscle development of preschool children. The purpose of this study is to investigate whether there is a relationship between PA levels and skeletal muscle thickness in preschoolers. By exploring this relationship, we hope to gain a better understanding of how PA can be used to promote healthy skeletal muscle development in preschoolers. Methods: In this study, a total of 275 healthy Japanese preschoolers, aged 4–6 years, from seven nursery schools in the town of Togo were recruited. Participants were asked to wear an accelerometer for four consecutive days to record their daily steps and the amount of time spent in moderate-to-vigorous PA and t total physical activity. Muscle thickness (MTs) was measured using B-mode ultrasonography at four sites: the anterior and posterior thigh (AT and PT, respectively) and the anterior and posterior lower leg (AL and PL, respectively). Results: On weekdays, boys were found to be more physically active and engaged in significantly higher levels of total physical activity and moderate-to-vigorous PA than girls. Both boys and girls recorded more physical activity, daily steps, and higher levels of total physical activity and MVPA on weekdays compared to weekends. After adjusting for daylight duration, multivariable regression analyses revealed that increased total physical activity and moderate-to-vigorous PA were positively associated with greater muscle thickness size in the anterior tibialis (AT) and posterior lower leg (PL) muscles (β = 1.11 and β = 1.37 for AT, β = 1.18 and β = 0.94 for PL, *p* < 0.05) in Japanese preschoolers. Conclusions: The time spent involved in most of the different categories of moderate-to-vigorous PA was significantly higher for boys than for girls on the weekdays and weekends. Additionally, there was a positive correlation between time spent in moderate-to-vigorous PA and greater development of skeletal muscle in the lower body.

## 1. Introduction

It is increasingly acknowledged that physical activity (PA) and physical inactivity are two distinct concepts that can have significant impacts on body composition and health-related quality of life [1]. PA is defined as any movement that results in energy expenditure, such as exercise, sports, and leisure activities. On the other hand, physical inactivity refers to a lack of bodily movement or a sedentary lifestyle, which includes sitting or lying down for extended periods of time without engaging in any PA. It is important to distinguish between the two concepts, as physical inactivity has been linked to negative health outcomes, such as obesity, cardiovascular disease, and metabolic disorders [2,3]. The absence of PA directly influences metabolism, bone mineral content [4], vascular health, and overall cardiometabolic dysfunction, with mechanisms biologically distinct from those associated with the cardiometabolic benefits of PA [5]. PA has independent and qualitatively different effects on human metabolism, such as muscle mass, fat mass [6], and body composition [7], and has been linked with adult hypertension, type 2 diabetes, and their premorbid risk factors. Moreover, recent research indicates that PA declines with aging during adolescence and was especially pronounced in girls and on weekends [8], although evidence on the magnitude of the decline is equivocal. In addition, inactivity may also continue throughout adulthood [9], increasing the risk of health issues in later life [10]. Thus, by assessing PA levels at an early age, parents and caregivers can identify areas where improvements can be made and implement strategies to encourage PA and prevent physical inactivity.

Muscle thickness (MT) is often considered a surrogate measure for skeletal muscle mass. Two large muscle groups are located in the thigh: the quadriceps femoris muscles in the anterior of the thigh that play an important role in performing basic PA such as walking and running, and the hamstring in the posterior that plays an important role in controlling lower body movements such as knee flexion and hip extension. During the last several decades, the quadriceps muscle group has been commonly used as a marker of age-related changes in muscle size and lower limb strength [11,12]. The maintenance of skeletal muscle throughout a life span is important from both a functional and metabolic perspective [13]. Skeletal muscle can react to a range of stimuli and is quite malleable, and it is an important predictor of several physiological capacities expressed in absolute terms [14]. Thus, there are many important PA interrelationships between health and aging that may contribute to the age-related muscle mass decrease in middle- and older-aged populations. Scott et al. [15] reported that ambulatory activity is correlated with both leg strength and leg muscle quality in women aged 50–79 years. The relationship between ambulatory activity and both leg strength and leg muscle quality were nonsignificant in men. Moreover, studies on the relationship between PA and skeletal muscle mass have focused mainly on elderly people, who have less physical inactivity and decreased muscle mass due to the natural physiological process of aging [16]. Gender, body composition, and pubertal state are essential factors that impact adolescent muscular strength and PA levels. Engaging in regular PA can lead to improvements in body composition, specifically an increase in fat-free mass, resulting in greater muscular strength. Understanding these factors can help promote PA and encourage healthy lifestyle behaviors in adolescents. [17,18]. It has been shown that the development of muscle thickness in the posterior thigh and lower leg of children before puberty is an important factor in promoting both the quantitative development of leg muscles and the development of motor ability [19]. Although numerous studies have investigated how age and gender impact skeletal muscles [20,21,22,23], the relationship between the amount and quality of PA and the distribution of skeletal muscle mass has not been extensively explored, particularly among preschool children. These children are a distinct population, as they are in a crucial developmental stage and are only beginning to establish their PA behaviors and routines. Therefore, it is crucial to understand the impact of PA on skeletal muscle development in this age group. Understanding the interrelationships among PA and skeletal muscle may be useful in the development of therapeutic strategies designed to increase skeletal muscle and to improve functional capacity for preschoolers.

Therefore, this study aimed to explore the potential association between the level of PA and the thickness of skeletal muscle in preschool children. By analyzing this relationship, we sought to gain a deeper understanding of how PA impacts skeletal muscle development in this critical developmental stage. Our hypothesis was that there would be a positive correlation between muscle mass and habitual PA (number of daily steps and duration of moderate-to-vigorous PA) in our assessments. To the best of our knowledge, this is the first study investigating the association between skeletal muscle and PA in Japanese preschoolers.

## 2. Methods

### 2.1. Participants

A total of 275 healthy preschoolers between the aged of 4–6 year from seven nursery schools in the town of Togo participated in the study. To participate in our study, children were required to meet several criteria, including regular attendance at the preschool that was under research, parental consent for their child to participate, and a lack of existing injuries or illnesses that could affect their PA levels or muscle development. By ensuring that all participants met these requirements, we were able to control potential confounding factors that could affect our results and obtain a more accurate understanding of the relationship between PA and muscle development in preschool-aged children. Parents/guardians of the children provided informed consent to participate in this study, which adheres to the guidelines of the Declaration of Helsinki and was approved by the Ethics Committee of Juntendo University.

### 2.2. Anthropometrics Data

Body mass, wearing only minimal clothing, was measured on a digital balance to the nearest 0.1 kg, and height was measured on a stadiometer to the nearest 0.1 cm. BMI (kg/m^2^) was calculated as body weight/height squared.

### 2.3. Muscle Thickness

B-mode ultrasonography was used to measure MT with a 5–18 MHz scanning head (Noblus; Hitachi, Tokyo, Japan). A water-soluble transmission gel was used to prepare the scanning head, allowing for acoustic contact without depressing the skin surface. The muscle thickness (MTs) was obtained at four different sites for both the anterior and posterior surfaces of the body. These sites were selected to represent the maximum muscle thickness of the target muscles and included locations on the thigh, lower leg, upper arm, and forearm. Measuring muscle thickness at these sites allowed us to gain a more comprehensive understanding of muscle development and changes in muscle mass throughout the body. The sites included the anterior and posterior thigh, and the anterior and posterior lower leg. The anatomical landmarks for the chosen sites were defined as follows: anterior (AT) and posterior (PT) thighs were on the anterior and posterior surfaces of the upper leg, respectively, midway between the lateral condyle of the femur near the knee and the greater trochanter at the hip; anterior (AL) and posterior (PL) lower leg muscles were measured at a proximal location approximately 30% between the lateral malleolus of the tibia near the knee and the lateral malleolus of the fibula near the ankle. This specific location was chosen to accurately capture changes in muscle size in the lower leg, which is particularly important for understanding age-related changes in muscle mass and function.

### 2.4. Physical Activity

We measured PA using a uniaxial accelerometer (AC), the Kenz Lifecorder GS (Suzuken Co., Ltd., Nagoya, Japan; weight 60 g). Preschoolers attached the AC to their waists and wore it from the time they woke up until bedtime. They were instructed to wear the AC while providing data for consecutive days. The accelerometer had previously been proven reliable in children and adolescents [24,25]. Intensity levels were classified as light (LPA, AC intensity levels of 1–3 and 1.5–2.9 METs), moderate (MPA, AC intensity levels of 4–6 and 3.0–5.9 METs), and vigorous (VPA, AC intensity levels of 7–9 and ≥6.0 METs) [26,27]. We recorded crude step counts to estimate activity levels, and the time spent in moderate-to-vigorous PA (≥3.0 METs) was calculated as the sum of the MPA and VPA minutes. Total PA (TPA) was calculated as the daily sum of LPA, MPA, and VPA. For inclusion in the following study, there has to be a minimum of 10 h of wear time per day over the course of at least two days of recording (including a weekend day) [28]. Furthermore, we excluded days during which no signal was detected for more than one hour, which was considered as non-wearing time. This approach was used to ensure that the data accurately reflected the actual PA levels of the participants and to avoid the inclusion of data from periods when the accelerometer was not worn.

### 2.5. Statistical Analyses

All data were analyzed using SPSS Statistics Version 22.0. (IBM Corporation, Chicago, IL, USA). For descriptive analyses, the results are presented as the mean ± SD. Gender differences in BMI and MT variables were analyzed using independent *t*-tests. All data were tested for normality using the Shapiro–Wilk test. Nonnormally distributed variables of nonparametric tests were used to evaluate groupwise differences. The effects of sex and day of the week on PA outcomes were examined using two-way ANOVA models. An ANOVA was utilized with the Bonferroni post hoc test to establish significance. Multivariate linear and linearity-using residual plots were used to investigate the relationships between PA and muscle size. Each variable of PA behavior per week (daily steps, Total PA, light PA, and moderate-to-vigorous PA) and muscle size (AT, PT, AL, PL) was subjected to regression analyses (AT, PT, AL, and PL). Variables that were reported and objectively measured were examined and presented individually. Our visually assessed the linearity between the residuals and dependent variables in the model and checked for normal distribution and homogeneity of variance of the residuals. Covariates included age, height, weight, and PA monitor wear time. We adjusted for age, height, and weight to the original model (Model 1). The second model was modified to account for daylight hour (Model 2). Thus, the daily step, Total PA, light PA, and moderate-to-vigorous PA per week were calculated as [(weekday average values ∗ 5) + (weekend average values ∗ 2)]/7. Boys and girls were included in the analyses since there was no moderating effect of gender on the relationship between PA and muscle size. Standardized parameters, *p*-values and R2 increase are presented. The significance level was set at *p* < 0.05 for all statistical analyses.

## 3. Results

Characteristics of the 275 participants are shown in Table 1. The sample was 45.1% boys, and the mean age of participants was 5.5 ± 0.6 years. Independent *t*-tests revealed no significant gender differences in BMI and MT variables. Boys engaged in higher levels of PA than girls on weekdays, including more Total PA and moderate-to-vigorous PA (*p* < 0.05). Furthermore, compared with weekends, weekdays showed more PA and significantly more daily steps, Total PA, light PA, and moderate-to-vigorous PA in both boys and girls.

Table 2 presents the outcomes of models 1 and 2 for variables that were assessed objectively and subjectively, respectively. Adjusting for height, weight, and daylight duration had little to no effect on the regression coefficients (Model 2).

When classified in PA intensities and based on model 2, we found a positive relationship between PA and muscle size. Our fundings suggest that AT’s muscle size would increase with a daily increase in Total PA (β = 1.11, *p* < 0.05) and PL (β = 0.94, *p* < 0.05) for preschool children. The correlation between muscle size in AT and PL increased with moderate-to-vigorous PA at the same rate as the positive correlation between Total PA and muscle size in AT (β = 1.37, *p* < 0.05) and PL (β = 1.18, *p* < 0.05) for preschool children. However, the number of daily steps and LPA showed no clear correlation with muscle size for preschool children, respectively.

## 4. Discussion

Preschool children are always subject to dynamic physical growth and development. Due to the significant impacts of PA on the health of preschool children, an accurate understanding of the association between PA and skeletal muscle size of great importance. The results indicate that, over the course of a week, higher mean levels of moderate-to-vigorous PA are correlated with thicker muscle size in the thigh and lower leg. This is not surprising, given that the muscles in the lower body are required for most common actives (i.e., walking, running, stair climbing). In addition, our results indicate that skeletal muscle size development can be promoted by moderate-to-vigorous PA for preschool children.

The relationship of PA with muscle size is poorly understood especially in preschool children. Interestingly, a recent study observed positive correlations between vigorous PA and lower-body muscle strength in tests of adolescents and adults [11,29]. The preschool children who participated in resistance training had significantly higher muscular strength scores than their peers who engaged in low or moderate levels of PA. However, when comparing resistance trainers to non-lifters who were in the highest tertile of PA, there was no significant difference in muscular strength. These findings suggest that resistance training may be an effective means of promoting muscular strength in preschool children, even in those who are highly active through other types of PA. Moreover, it highlights the importance of incorporating resistance training into PA programs for preschool children in order to promote optimal muscular strength development [30]. Additionally, research has indicated that as individuals age, both men and women tend to experience a decline in skeletal muscle mass that is specific to certain areas of the body. In particular, there is a notable reduction of muscle mass in the muscles located in the anterior upper leg region. This site-specific loss of skeletal muscle mass can have significant implications for overall health and mobility, highlighting the importance of promoting PA that targets these muscle groups as individuals age [31]. Furthermore, understanding the factors that contribute to this age-related decline in skeletal muscle mass can help inform interventions that promote healthy aging and prevent the onset of chronic conditions associated with muscle loss. In addition, a longitudinal study has shown that there are significant declines in the cross-sectional area of the quadriceps muscle over time, particularly as individuals age. Interestingly, no changes were observed in the muscle mass of the posterior thigh over the same period of time. These findings suggest that different muscle groups may be affected differently by the aging process and highlight the importance of targeted PA interventions to maintain muscle mass in areas that are particularly vulnerable to age-related decline [32]. The results of the present study and the previous studies together suggest that the duration of PA is an important predictor of, and may be strongly associated with age-related, site-specific increase of thigh muscle.

The distinct natures of PA and muscle size include physiological responses and adaptations that are not merely opposites of one another. However, there is a lack of evidence to determine the optimal amount of PA for an adequate skeletal development in preschoolers. Our study shows that moderate-to-vigorous PA was positively correlated with muscle size in the thigh and lower leg, with a stronger association observed in boys. Furthermore, a recent study has provided evidence that increasing moderate PA or vigorous PA by 10 min per day can lead to an average increase of 1–2% in bone stiffness based on a large sample of children aged 2–10 years [33]. Park et al. [34] demonstrated that muscle mass in the lower extremities as measured by whole-body dual-energy X-ray absorptiometry was associated with PA such as the daily step count based on a sample of children aged 2–3 years. Katzmarzyk et al. analyzed 356 boys and 284 girls (9–18 years) from the Quebec Family Study and found a weak association between PA and the static strength of the legs in a maximal voluntary isometric contraction at a knee angle of 90° and a low/no significant correlation between PA and the sum of skinfolds [35]. In other words, it can be difficult to determine whether the observed changes in muscle size and strength are solely due to PA or if other factors such as genetics or growth and development also play a role. However, studies have shown that engaging in regular PA, particularly moderate-to-vigorous PA, can have significant benefits for muscle size and strength, as well as bone health and overall physical fitness. Additionally, promoting PA in children and adolescents can have long-term health benefits and help establish healthy habits that can be carried into adulthood. Furthermore, Japan’s sports policy prioritizes the promotion of sports participation throughout one’s life and the development of appropriate PA habits from an early age [36]. The research results provide valuable baseline data for improving PA among children, and by cultivating a strong foundation of physical fitness in children, contribute to the promotion of sports as a means of achieving this goal.

In this study, the distribution of muscle mass differed between boys and girls. In individuals with similar body height and weight, boys had larger lower- and smaller upper-extremity muscles as compared to girls. These differences may partly be caused by the fact that most boys performed PA, predominantly doing more moderate-to-vigorous PA, and the kind of PA may play an important role for MT. For example, previous studies reported a consistent mean increase of 3–4% in thigh muscle volume following a 5-week training program in boys and girls [37,38]. However, it is important to consider that most children do not engage in highly structured training regimens such as those imposed in exercise training studies; therefore, the importance of considering the influence of habitual, free-living PA over a period of time (i.e., the preschoolers growth spurt) is a novel and practical aspect of the present study. Nevertheless, our findings suggest the influence on PA with thigh muscle tended to increase with age, both in boys and girls. This finding is important, since preschoolers represent not only the period of the lifespan when PA levels increase substantially [39,40], but also a time when substantial changes in body composition are occurring, so it becomes more important to ascertain the positive influence of habitual PA. Furthermore, the development of skeletal mass has important implications for metabolic health [41].

There are some limitations of this study. First, since our sample was only in urban areas, our findings may not apply to preschoolers in other regions of Japan. Secondly, the cross-sectional design makes it impossible to establish a causal relationship between high PA and greater skeletal muscle mass content. Thirdly, accelerometers also have some limitations with respect to assessing the overall activity levels. It is important to note that, for children, a shorter accelerometer period (e.g., 5 s) is highly recommended due to the intermittent nature of their PA [42]. Consequently, it is possible that the amount of sedentary time and vigorous PA was underestimated, while moderate PA may have been overestimated as a result of the shorter accelerometer period recommended for children. This consideration should be taken into account when interpreting data related to children’s PA levels. Finally, we acknowledge the limitation of a cross-sectional design and recognize that longitudinal studies are needed to better understand the associated between PA and MT of children from an early age.

## 5. Conclusions

In conclusion, this study examined the relationship between preschool children’s skeletal muscle development and PA by objective measurements and whether their gender made a difference. Furthermore, the data indicates that boys spent significantly more time in most moderate-to-vigorous PA than girls on both weekdays and weekends. Additionally, higher levels of moderate-to-vigorous PA were found to be positively associated with greater skeletal muscle mass in the lower body. The relationship of PA and skeletal muscle should play a greater role in questions of obesity and osteoporosis prevention undertaken during childhood.

## Figures and Tables

**Table 1 children-10-00455-t001:** Descriptive statistics for characteristics, muscle thickness, and physical activity in both boys and girls.

	All	Boys	Girls	Sex Difference	Daily Difference
	*n* = 275	*n* = 124	*n* = 151		
Age	5.5 ± 0.5	5.5 ± 0.6	5.5 ± 0.5		
Height (cm)	109.5 ± 5.0	109.7 ± 5.0	109.4 ± 5.0	*p* = 0.25	
Weight (kg)	18.4 ± 2.7	18.3 ± 2.6	18.4 ± 2.8	*p* = 0.16	
BMI (kg/m^2^) ^a^	15.2 ± 1.5	15.2 ± 1.4	15.3 ± 1.5	*p* = 0.12	
Muscle Thickness ^a^					
AT (mm)	24.1 ± 2.7	23.6 ± 2.6	24.4 ± 2.8	*p* = 0.16	
PT (mm)	33.5 ± 2.9	34.0 ± 3.0	33.0 ± 2.8	*p* = 0.10	
AL (mm)	13.3 ± 1.3	13.6 ± 1.2	13.2 ± 1.4	*p* = 0.22	
PL (mm)	37.9 ± 3.0	38.1 ± 2.8	37.7 ± 3.2	*p* = 0.19	
Physical activity on weekdays ^b^					
Daylight duration (hour/day)	11.1 ± 1.3	10.9 ± 2.1	11.2 ± 1.5	*p* = 0.33	
Daily step (step)	15,373 ± 3392	16,206 ± 3546	16,488 ± 3108	*p* = 0.16	
TPA (min)	153.8 ± 33.0	161.8 ± 33.8	147.2 ± 31.0	*p* = 0.07	
LPA (min)	102.6 ± 20.5	106.8 ± 20.4	99.1 ± 20.0	*p* < 0.05	
MVPA (time)	51.2 ± 15.9	55.0 ± 17.4	48.0 ± 13.8	*p* < 0.05	
Physical activity on weekends ^b^					
Daylight duration (hour/day)	10.5 ± 1.1	10.3 ± 1.2	10.5 ± 1.0	*p* = 0.29	*p* = 0.35
Daily step (step)	10,629 ± 4016	10,514 ± 3913	10,724 ± 4110	*p* = 0.16	*p* < 0.05
TPA (time)	109.0 ± 40.0	108.6 ± 39.7	109.3 ± 40.3	*p* = 0.07	*p* < 0.05
LPA (time)	77.3 ± 26.3	77.4 ± 26.3	77.2 ± 26.4	*p* = 0.31	*p* < 0.05
MVPA (time)	32.2 ± 16.2	31.2 ± 16.2	32.1 ± 11.7	*p* = 0.25	*p* < 0.05

Data are present as mean ± SD. BMI: body mass index; AT: anterior thigh; PT: posterior thigh; AL: anterior lower leg; PL: posterior lower leg; TPA: total physical activity; LPA: light physical activity; MVPA: moderate to vigorous physical activity. ^a^ Tested with *t*-test. ^b^ Tests with Bonferroni-corrected analyses of variance.

**Table 2 children-10-00455-t002:** Multivariate linear regression investigating the association of accelerometer-based PA data with muscle size.

	Model 1 ^a^			Model 2 ^b^		
	Adjusted for Sex, Age, Height and Weight			Model 1 + Adjusted for Daylight Duration		
	β	*p* Value	R^2^	β	*p* Value	R^2^
Anterior thigh						
Daily steps (steps/day)	0.11	0.15	0.151	−0.84	0.75	0.169
LPA (min/day)	0.13	0.21	0.175	−0.05	0.89	0.183
MVPA (min/day)	0.76	<0.05	0.181	1.37	<0.05	0.242
TPA (min/day)	1.23	<0.05	0.177	1.11	<0.05	0.238
Posterior thigh						
Daily steps (steps/day)	1.86	0.87	0.156	1.02	0.86	0.157
LPA (min/day)	0.68	0.96	0.227	0.43	0.15	0.225
MVPA (min/day)	0.55	0.93	0.238	1.74	0.98	0.263
TPA (min/day)	−0.91	0.74	0.167	0.94	0.26	0.267
Anterior lower leg						
Daily steps (steps/day)	0.72	0.56	0.208	−1.33	0.45	0.296
LPA (min/day)	0.45	0.38	0.342	0.09	0.69	0.263
MVPA (min/day)	0.01	0.92	0.174	−0.44	0.11	0.272
TPA (min/day)	0.05	0.22	0.237	−0.04	0.50	0.184
Posterior lower leg						
Daily steps (steps/day)	0.72	0.56	0.208	−0.03	0.45	0.196
LPA (min/day)	0.45	0.38	0.142	−0.01	0.35	0.242
MVPA (min/day)	1.61	<0.05	0.248	1.18	<0.05	0.190
TPA (min/day)	0.81	<0.05	0.246	0.94	<0.05	0.186

TPA: total physical activity; LPA: light physical activity; MVPA: moderate to vigorous physical activity. The standardized regression coefficient (β), *p* value, and coefficient of determination (R^2^) are given for each association. PA: physical activity. ^a^ In Model 1, the independent variables were adjusted as follows: sex, age, height, and weight. ^b^ In Model 2, the independent variables were (in addition to Model 2) adjusted as follows: daily steps and time of accelerometer based MVPA and TPA per week.

## Data Availability

The data presented in this study are available on request from the corresponding author. The data are not publicly available due to privacy restrictions.

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
