# Peer review of "Relationship between Skeletal Muscle Thickness and Physical Activity in 4- to 6-Year-Olds in Japan"

_children, 2023, doi:10.3390/children10030455_

Round 1

Reviewer 1 Report

Nice compact study.  Presented very nicely.  I personally feel that this article should be available to pre K-3 educators, specifically physical educators.

Author Response

Dear Reviewer,

I will send you the revised paper.

I would like to express my gratitude to you for taking the time to review my manuscript. I appreciate the careful consideration and useful comments provided.

Despite your busy schedule, thank you very much for your valuable feedback.

Best regards,

PENGYU DENG

Reviewer 2 Report

The manuscript “Relationship between skeletal muscle and physical activity in 4- to 6-years olds” aimed to investigate the association between muscle thickness and levels of physical activity in infants. I commend the authors for their work in preparing the manuscript. General comments and specific points and sections are provided below:

General comments

Title:

- The title should state muscle thickness, not only muscle.

Abstract:

- The abstract is well-written and concise. Is clearly describes the design of the study and provides the obtained data and conclusions.

Introduction:

- Overall, the introduction is well-written.

- The authors successfully point the importance of physical activity during childhood but fail to provide the rationale for investigating muscle thickness in this population. Why is it important for children aged 4- to 6-years to have thick lower-limb muscles? The authors do present data from older women, which is relevant to their metabolic and functional status, but can this be extrapolated to preschool children?

Materials and Method:

- The sample size is considerable, and I commend the authors for that.

- The statistical procedures are adequate, however some of the analyses are supported by the rationale provided in the introduction. If the authors choose to maintain all analyses, please include the relevance of investigating the effects of sex and day of the week on physical activity levels in the introduction.

Results:

- The results are adequately presented. Please find specific comments at the end of this review sheet.

Discussion:

- The discussion is succinct and based off the main findings of the study. It successfully contrast the obtained findings with the yet scarce literature on the topic. I commend the authors for the discussion.

Conclusion:

- The conclusion is short and concise, as should be.

Please find specific comments detailed below:

L50: anterior and posterior compartments of the thigh

L51: are quadriceps muscles really important for lateral rotation of the leg?

Table 1: Please remove the lines between variables. I would recommend the use of lines to discriminate groups of variables (i.e., anthropometric, muscle thickness, PA on weekdays and PA on weekends) only.

Table 2: The same comment for table 1 apply here.

Author Response

Dear Reviewer,

Despite your busy schedule, thank you very much for your valuable feedback.

We have studied comments carefully and have made correction which we hope meet with approval. We tried our best to improve the manuscript and made some changes in the manuscript. These changes will not influence the content and framework of the paper. Note:Red marks the revision.

1.- The title should state muscle thickness, not only muscle.

Response: We have made the necessary corrections to the title, as you pointed out.

2.- The authors successfully point the importance of physical activity during childhood but fail to provide the rationale for investigating muscle thickness in this population. Why is it important for children aged 4- to 6-years to have thick lower-limb muscles? The authors do present data from older women, which is relevant to their metabolic and functional status, but can this be extrapolated to preschool children?

Response: We have added previous research indicating that the development of muscle thickness in the lower limb plays an important role in the quantitative development of leg muscles and the development of motor ability (L36-39). 

3.- The statistical procedures are adequate, however some of the analyses are supported by the rationale provided in the introduction. If the authors choose to maintain all analyses, please include the relevance of investigating the effects of sex and day of the week on physical activity levels in the introduction. 

Response: We have expanded the investigation to include the effect of sex and day of the week on Physical activity in the introduction (L11).

4.- anterior and posterior compartments of the thigh are quadriceps muscles really important for lateral rotation of the leg?

Response: We have made the corrections you suggested regarding the description of the quadriceps muscle (L17-20).

5.-Table 1: Please remove the lines between variables. I would recommend the use of lines to discriminate groups of variables (i.e., anthropometric, muscle thickness, PA on weekdays and PA on weekends) only. 

Response: As you pointed out, we have made the necessary corrections to the table.

6.-Table 2: The same comment for table 1 apply here.

Response: As you pointed out, we have made the necessary corrections to the table.

Reviewer 3 Report

Thank you for sharing this paper with me. This is very interesting and important. I have several minor comments.

1. There are several important assumptions of ANOVA/OLS. Please describe how you have met these assumptions as part of the method section.

2. Are there any policy implications based on your findings? Are there any possibly useful intervention programs?

3. Please mention that this study was done in Japan in the title and abstract.

Author Response

Dear Reviewer,

Despite your busy schedule, thank you very much for your valuable feedback.

We have studied comments carefully and have made correction which we hope meet with approval. We tried our best to improve the manuscript and made some changes in the manuscript. These changes will not influence the content and framework of the paper. Note:Red marks the revision.

  1. There are several important assumptions of ANOVA/OLS. Please describe how you have met these assumptions as part of the method section.

Response: In the Method section, we have included normality, homoscedasticity, and linearity as assumptions of ANOVA/OLS (L91-93 and L99-101).

  1. Are there any policy implications based on your findings? Are there any possibly useful intervention programs?

Response: We have expanded the discussion section to include additional policy implications and potential intervention programs that could be developed based on our findings (L165-169).

Additionally, we have a researcher in our group who teaches physical education and provides guidance on children's physical activity in kindergartens. If the opportunity arises in the future, we would like to share our findings on this topic through this platform.

  1. Please mention that this study was done in Japan in the title and abstract.

Response: We have made the corrections to the title and abstract as you pointed out.
